**communications** engineering

# Scalable reinforcement learning for large-scale coordination of electric vehicles using graph neural networks

Stavros Orfanoudakis [1] ✉, Valentin Robu [2,3], E. Mauricio Salazar[3], Peter Palensky[1] & Pedro P. Vergara[1]

As the adoption of electric vehicles (EVs) accelerates, addressing the challenges of large-scale, city-wide optimization becomes critical in ensuring efficient use of charging infrastructure and maintaining electrical grid stability. This study introduces EV-GNN, a novel graph-based solution that addresses scalability challenges and captures uncertainties in EV behavior from a Charging Point Operator's (CPO) perspective. We prove that EV-GNN enhances classic Reinforcement Learning (RL) algorithms' scalability and sample efficiency by combining an end-to-end Graph Neural Network (GNN) architecture with RL and employing a branch pruning technique. We further demonstrate that the proposed architecture's flexibility allows it to be combined with most state-of-the-art deep RL algorithms to solve a wide range of problems, including those with continuous, multi-discrete, and discrete action spaces. Extensive experimental evaluations show that EV-GNN significantly outperforms state-of-the-art RL algorithms in scalability and generalization across diverse EV charging scenarios, delivering notable improvements in both small- and large-scale problems.

Efficient utilization of Electric Vehicles (EVs) is crucial for complying with the current energy transition goals[1]. The adoption of EVs accelerates the challenges associated with maintaining a reliable and efficient charging network, which have become increasingly significant[2]. Ensuring the optimal use of existing charging infrastructure is essential to meet the demands of the growing number of EVs[3]. Addressing scalability concerns is essential to supporting the widespread adoption of EVs and preventing congestion in distribution networks[4,5]. Congestion in electricity networks leads to significant economic inefficiencies, higher energy prices, and increased operational costs, with total costs varying by country, energy market structures, and congestion severity. For instance, U.S. transmission congestion costs rose from $13 billion in 2021 to around $20 billion in 2022[6]. These rising costs underscore the need for Charge Point Operators (CPOs) to implement scalable solutions for efficiently dispatching large numbers of EVs, helping to alleviate distribution network congestion and minimize associated socioeconomic impacts[7].

Smart charging is a complex optimization task, often formulated as a mixed integer programming (MIP) problem with many decision variables, constraints, and uncertainties in energy generation, consumption, and EV arrival/departure times. Stochastic optimization approaches address EV behavior uncertainties, such as arrival times, to maximize grid load factors[8].

MIP models have been applied to offline and online EV scheduling, typically for single stations[9]. Multi-stage algorithms aim to minimize costs while balancing supply and demand[10], and event-triggered scheduling handles uncertainties but may struggle with real-time execution for many EVs[11]. Chance-constrained optimization improves efficiency using distributed algorithms such as the alternating direction method of multipliers[12]. When it comes to real-time decision-making, model predictive control (MPC) methods, including robust variants, have been employed for real-time scheduling of EVs, achieving near-optimal profit and meeting charging requests even under high uncertainty[13,14]. However, these approaches face challenges as the number of decision variables and constraints increases, a common issue in practical contexts where CPOs must frequently rerun optimization algorithms, typically every 5 minutes. This operational demand, especially with the anticipated adoption of EVs in the near future, presents a significant operational challenge.

Reinforcement Learning (RL)[15] is a powerful technique for solving sequential decision-making problems. RL algorithms offer particular advantages for the EV dispatch problem, such as non-linear modeling, better uncertainty quantification, and faster execution speed compared to traditional mathematical programming methods[16]. Classic RL approaches, which model problems as Markov Decision Processes (MDPs), have been

[1]Intelligent Electrical Power Grids, Delft University of Technology, Delft, The Netherlands. [2]Intelligent & Autonomous Systems Group, Centrum Wiskunde & Informatica (CWI), Amsterdam, The Netherlands. [3]Electrical Energy Systems, Eindhoven University of Technology (TU/e), Eindhoven, The Netherlands. ✉e-mail: s.orfanoudakis@tudelft.nl

successfully applied to decentralized decision-making for EV charging using algorithms, such as Deep Deterministic Policy Gradient (DDPG)[17], Soft-Actor-Critic (SAC)[18], and batch RL methods[19]. However, these methods often lack constraint satisfaction guarantees and struggle with scalability as state and action spaces grow[20,21]. Safe RL frameworks aim to address these limitations by formulating the problem as constrained MDPs, ensuring constraints are met but often at the cost of optimal performance and scalability[22,23]. Multiagent RL (MARL) approaches, such as those employing the centralized training decentralized execution paradigm, attempt to reduce state and action dimensionality by distributing decision-making across agents[24,25], where each agent could be a single charging point, a charging station, an aggregator, etc. Despite these efforts, MARL methods still face convergence challenges as the number of agents increases and issues with achieving optimality in complex environments[26]. To address these issues and better utilize graph-structured data, RL methods have been integrated with Graph Neural Networks (GNNs)[27]. GNNs have primarily been used as feature extractors in the EV charging context, particularly for EV charging station recommendation[28] due to the graph-like nature of transportation networks. However, to the best of our knowledge, no study has fully developed a comprehensive end-to-end RL GNN approach to leverage the unique properties of the GNN architecture for solving optimization problems[29–31] in the context of EV charging.

Through close collaboration with commercial CPOs in the Netherlands, it has become evident that the current charging approaches are inefficient when it comes to managing more than a few hundred EVs at a time. However, with the growing demand, they are now facing the challenge of handling thousands of EV charging requests simultaneously. In this study, we introduce EV-GNN, an innovative approach that combines a novel graph MDP formulation of the EV charging problem with dynamic end-to-end GNNs, enhancing the scalability features of state-of-the-art RL algorithms.

EV-GNN not only enhances the scalability of state-of-the-art RL algorithms but also empowers them to handle the complexity and scale of real-world EV charging scenarios. Unlike conventional methods such as stochastic optimization, MPC, or traditional RL approaches, which often struggle with large-scale problem instances, EV-GNN enables efficient and robust decision-making, opening the door to practical applications in large-scale environments. Our approach applies to any optimal EV charging problem from the perspective of a CPO that manages charging stations. We prove in this study, that EV-GNN significantly enhances classic RL algorithms' scalability and sample efficiency. We demonstrate with extensive experimental evaluations the improved capabilities of our approach over state-of-the-art RL algorithms across various EV charging scenarios, including continuous, and multi-discrete action spaces. Our approach shows significant performance improvements in both small and large-scale optimal EV charging problems, which is possible due to the proposed end-to-end GNN architecture. Furthermore, we provide explainability experiments that illuminate the mechanisms behind our end-to-end RL GNN approach's enhanced sample efficiency, offering valuable insights into the workings of our novel approach.

## Results
### RL for optimal EV charging
Developing scalable RL methods is crucial for efficiently dispatching large numbers of EVs and preventing congestion in distribution networks[32]. With the rise of distributed energy resources (DERs) such as photovoltaic (PV) systems, wind turbines, and EVs, grid control has become increasingly dynamic. The presence of uncertain components requires efficient energy distribution to all connected loads. Power setpoint tracking (PST) is a key challenge for CPOs, involving the fair and efficient allocation of grid capacity to EVs while minimizing the gap between available and actual charging power in real-time. This real-time decision-making, occurring usually every 5 minutes with limited information, is complicated by DER uncertainties and energy market volatility. Inaccurate power tracking can result in significant costs for CPOs and market operators.

The PST problem is divided into discrete time periods $\mathcal{T}$, where each $t \in \mathcal{T}$. A fixed number of charging stations $\mathcal{C}$, connected to a transformer $w \in \mathcal{W}$, serve EVs that dynamically connect to charging points (CPs), denoted by $j \in \mathcal{J}$, at charger $i$. The goal is to align actual power usage ($P^{\text{tot}}$) with procured power ($P^{\text{set}}$) in real-time, minimizing the cumulative PST error:

$$\min \sum_{t \in \mathcal{T}} \left( P_t^{\text{set}} - P_t^{\text{tot}} \right), \tag{1}$$

while ensuring transformer safety, voltage stability, and reliable power supply.

Centralized EV charging optimization is often modeled as a single-agent learning task, where the CPO acts as the agent, and the state space $\mathcal{S}$ grows linearly with the number of charging points. Typically, this state is represented by a long vector containing similar variables, such as the state of charge (SoC) for each connected EV, and zeros for unoccupied points. However, such long vectors make it difficult to extract useful features using traditional neural networks such as Multi-Layered Perceptrons (MLPs).

The learning process of the RL agent is further complicated by the dynamic nature of EV arrivals and departures, leaving many charging points unoccupied for long periods[33]. Classic RL formulations assume a fixed-size action space $\mathcal{A} \in \mathbb{R}^{|\mathcal{J}|}$, leading to computing actions for unoccupied CPs. While manageable in small-scale cases, this issue significantly impedes RL performance in larger-scale settings due to invalid actions[34]. Addressing these challenges is essential for improving the scalability and efficiency of RL-based EV charging solutions.

### End-to-end RL GNN architecture
Optimal EV charging problems can be visualized as a collection of charging stations, either grouped together or directly connected to the electrical grid via a transformer. EVs arrive and depart based on drivers' preferences, such as overnight home charging or public charging while shopping. This setup can be formalized into a mathematical programming (See Section ?? for more details) problem where the CPO controls each charging station's (dis) charging power to minimize the PST error, reduce load peaks, maximize profits, etc. At the same time, the optimal EV charging problem is subject to constraints such as maximum power limits and ensuring that EVs reach a desired SoC by their departure time.

To efficiently handle the challenges of dynamic EV charging, a graph-based state-action representation is introduced, as shown in Fig. 1a. Without this graph structure, managing the complexity of dynamic state representation would be inefficient and less scalable. The problem is modeled using a graph-based approach where EVs and chargers are organized at the transformer level, managed by a CPO. In this graph, EV nodes connect to charger nodes via CPs, and charger nodes link to transformer nodes, representing grid connections. To improve RL training, branches not leading to EV leaf nodes are pruned, simplifying the graph and focusing on valid actions, with only relevant connections retained in the pruned graph $\mathcal{G} = (N, \mathcal{E})$.

Graph $\mathcal{G}$ consists of $N$ nodes, including $N^{\text{ev}}$ EV nodes with features $X^{\text{ev}} \in \mathbb{R}^{N^{\text{ev}} \times F^{\text{ev}}}$, $N^{\text{cs}}$ charger nodes with $X^{\text{cs}} \in \mathbb{R}^{N^{\text{cs}} \times F^{\text{cs}}}$, $N^{\text{tr}}$ transformer nodes with $X^{\text{tr}} \in \mathbb{R}^{N^{\text{tr}} \times F^{\text{tr}}}$, and $N^{\text{cpo}}$ CPO nodes with $X^{\text{cpo}} \in \mathbb{R}^{N^{\text{cpo}} \times F^{\text{cpo}}}$. As shown in Fig. 1b, the GNN feature extractor processes nodes by type using node-type-specific MLPs. Each node type $p \in \{\text{ev, cs, tr, CPO}\}$ has its raw features $X^{(p)}$ transformed into embeddings $\widetilde{X}^{(p)}$ via:

$$\widetilde{X}^{(p)} = \sigma\left( W^{(p)} \cdot X^{(p)} + b^{(p)} \right), \tag{2}$$

where $\sigma(.)$ is the ReLU activation function, $W^{(p)} \in \mathbb{R}^{F^{(p)} \times F_0}$ and $b^{(p)}$ are trainable parameters. This results in a homogenized graph feature matrix $X_0 \in \mathbb{R}^{N \times F_0}$, where $F_0$ is the feature dimension. Then, the actor NN (Fig. 1c) processes the homogenized graph by applying $L$ layers of a Graph Convolutional Network (GCN)[27]. GCNs operate directly on the graph structure, leveraging the connectivity and relationships between nodes.

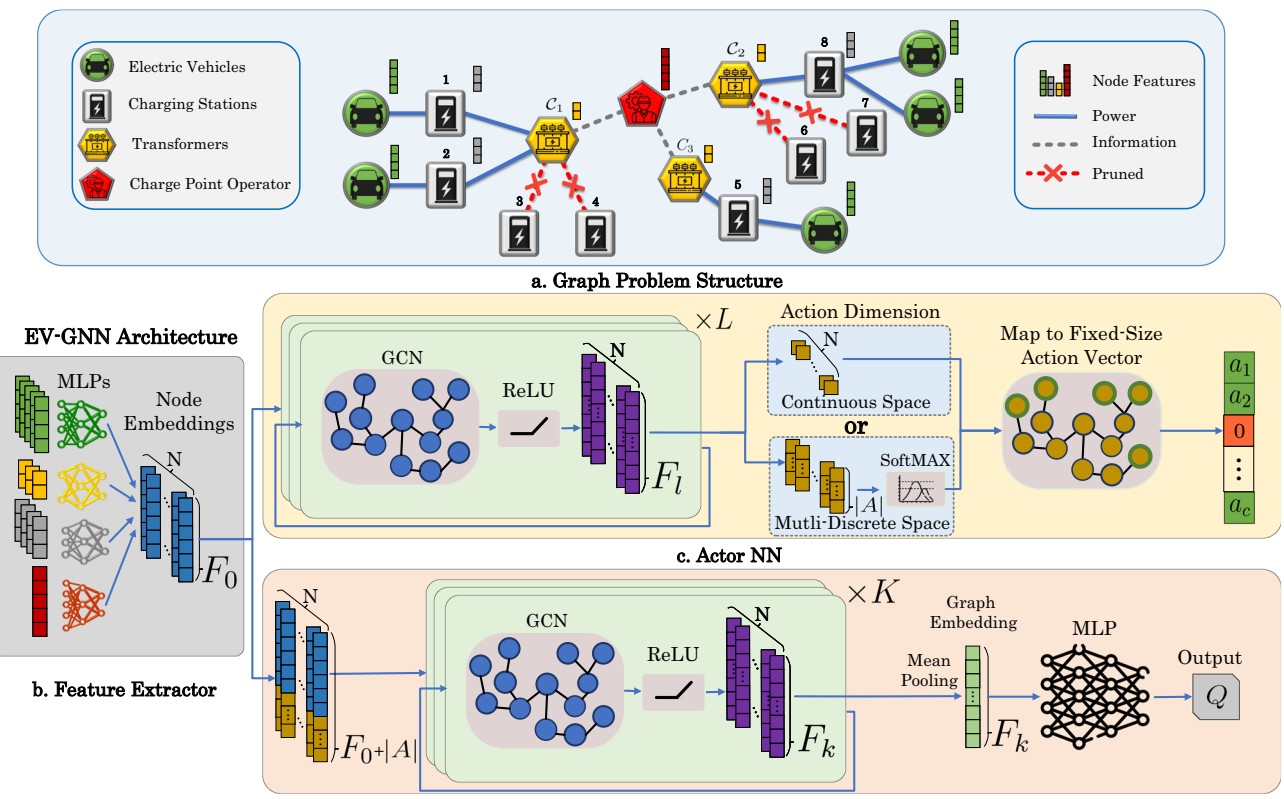

**Fig. 1 | The concept and architecture of EV-GNN. a** The EV charging optimization problem is modeled as a graph, with nodes representing components (EV, charger, transformer, CPO) and their unique features. The graph is simplified by pruning branches where no actions occur, such as charging stations 3, 4, 6, and 7 with no connected EVs. **b** Heterogeneous node features are processed through node-specific MLPs to transform them into higher-dimensional, homogeneous node embeddings of the same size $F_0$. **c** The actor NN consists of $L$ sequential GCN layers with $F_l$ features, which process the graph and reduce each node's feature to 1 for continuous actions or $|\mathcal{A}|$ for multi-discrete actions. EV node features are then selected and mapped to a fixed-size action vector representing the power injection or absorption. **d** The critic NN concatenates the actor's action features with the node state features, processes them through $K$ sequential GCN layers with $F_k$ features, and then mean pools the high-dimensional node features into a fixed-size graph embedding. This embedding is finally fed into an MLP to compute the Q-value.

GCNs perform message-passing among the nodes, enabling the aggregation of information from neighboring nodes to update each node's representation defined as:

$$X_l = \sigma\left(D^{-1/2}SD^{-1/2}X_{l-1}\Theta_l\right), \quad l = 1, \ldots, L \quad (3)$$

where $S$ is the adjacency matrix of the graph $\mathcal{G}$, and $D$ is the diagonal degree matrix with $D_{ii} = \sum_{j=0} A_{ij}$ for normalization. The learnable weight matrices of GCN layer $l$ are represented by $\Theta_l \in \mathbb{R}^{F_{l-1} \times F_l}$ and $\sigma$ is a nonlinear activation function, such as ReLU, that enables the approximation of complex problem dynamics.

The EV-GNN approach supports both continuous and (multi)-discrete RL problems. For continuous actions, the actor's last GCN layer has $F_L = 1$, and for multi-discrete actions, it has $F_L = |\mathcal{A}|$, followed by a Soft-MAX layer, where $|\mathcal{A}|$ is the number of discrete actions per node. For example, with $|\mathcal{A}| = 3$, an EV's actions could be 0 for *do-nothing*, 1 for *half-power charging*, and 2 for *maximum charging*. In the end, the EV node features $X_L$ are mapped to the action vector $\boldsymbol{a} \in \mathbb{R}^{|\mathcal{J}|}$, while pruned nodes take action 0 for no charging.

The critic network, shown in Fig. 1d, calculates the action-value function $Q(\boldsymbol{s}, \boldsymbol{a})$ by using the homogenized feature vector $X_0$ (state $\boldsymbol{s}$) concatenated with the actor's last GCN layer output $X_L$ (action $\boldsymbol{a}$). After $K$ sequential GCN layers, a global mean pooling operation converts node-level features $X_K \in \mathbb{R}^{N \times F_K}$ into a graph-wide embedding $\widetilde{X}_K \in \mathbb{R}^{F_K}$, ensuring symmetry and efficient learning of shared representations. The mean

pooling is defined as:

$$\widetilde{X}_K = \frac{1}{N}\sum_{n=1}^{N} X_{K,n}. \quad (4)$$

This graph embedding, invariant to the graph size $N$, is passed through an MLP to compute the Q-value, with the last layer outputting a single value.

The EV-GNN architecture handles MDPs with both static and dynamic state graphs and integrates seamlessly with various actor-critic RL algorithms, such as Twin Delayed DDPG[35] (TD3), Soft-Actor-Critic (SAC), and Proximal Policy Optimization[36] (PPO). Figure 1c–d illustrate the actor and critic networks for the DDPG and TD3 algorithms. Notably, the final GCN layer in the actor NN can be adapted from a single layer predicting actions to two parallel GCN layers-one predicting the mean and the other estimating the standard deviation of the action distribution, as seen in SAC and PPO. Minor adjustments to the actor and critic NN architectures enable the use of EV-GNN with the most state-of-the-art deep RL algorithms. Furthermore, depending on the specific problem and RL algorithm, the actor and critic networks may either share a feature extractor or use separate ones tailored to their respective functions.

**Making RL algorithms scalable**

Scalability, understood as the capability of an RL algorithm to handle large state and action spaces, is an important feature in the context of city-wide smart charging of EVs. Figure 2 demonstrates the need for a novel, scalable RL approach by comparing the maximum reward achieved across various experiment sizes (number of CPs) for the PST problem and a set of RL

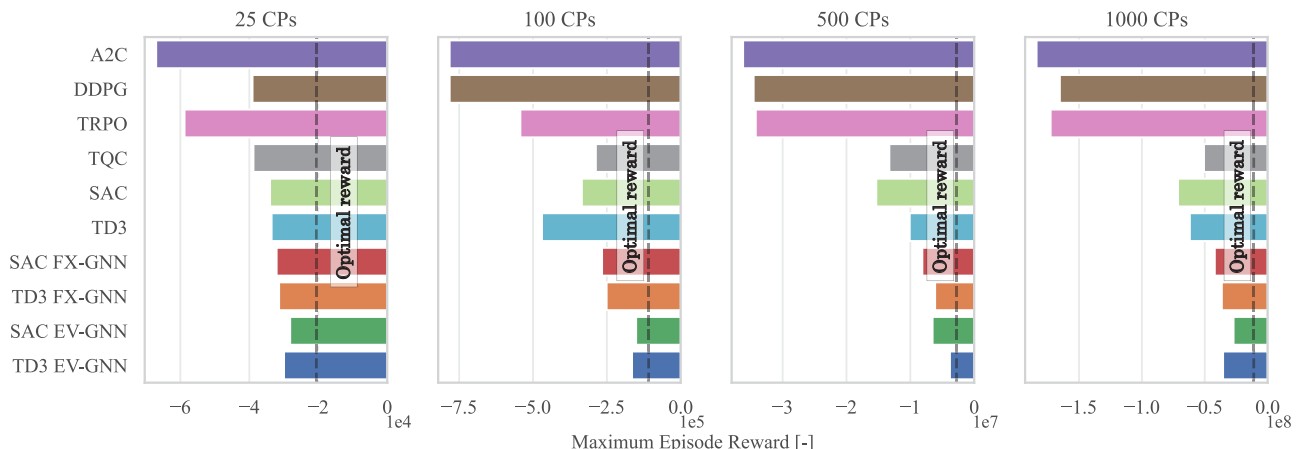

**Fig. 2 | Maximum episode reward obtained as a function of RL algorithm and experiment scale.** The scalability of classic state-of-the-art RL algorithms is compared with the proposed GNN-based approaches, demonstrating the limitations of traditional methods as the action and state spaces increase. The maximum reward achieved after 5 training runs with different seeds is presented. The results emphasize the superior performance of the EV-GNN versions, showcasing how significantly better they perform in comparison to the classic RL algorithms.

## Table 1 | Table of Abbreviations

| Abbreviation | Meaning | Abbreviation | Meaning |
|---|---|---|---|
| EV | Electric Vehicle | SAC | Soft Actor-Critic |
| CP | Charge Point | DDPG | Deep Deterministic Policy Gradient |
| CPO | Charge Point Operator | TD3 | Twin Delayed DDPG |
| PST | Power Setpoint Tracking Problem | TRPO | Trust Region Policy Optimization |
| AFAP | (charge) As Fast As Possible | PPO | Proximal Policy Optimization |
| RR | Round Robin | A2C | Advantage Actor-Critic |
| MPC | Model Predictive Control | TQC | Truncated Quantile Critics |
| RL | Reinforcement Learning | MLP | Multi-Layer Perceptron |
| MDP | Markov Decision Process | GNN | Graph Neural Network |
| GNN-FX | RL + GNN feature extractor | EV-GNN | RL + End-to-End GNN |

algorithms. Additionally, the optimal (oracle) reward is shown with a vertical line, which is obtained by solving the mathematical programming problem under the assumption that the model has knowledge of future events, which is impossible to achieve in real-time scenarios. Nevertheless, this comparison provides a measurement with respect to the theoretical optimal reward. In this study, four different experiment scales are defined based on the number of CPs a CPO controls. A smaller number of CPs (25) represents a simpler optimization task with fewer decision variables and constraints, while larger numbers (100, 500, 1000) present increasingly complex and challenging city-wide optimization problems that are difficult to solve in real-time.

Advantage Actor Critic[37] (A2C), DDPG, Trust Region Policy Optimization[38] (TRPO), Truncated Quantile Critics[39] (TQC), SAC, and TD3 are state-of-the-art RL algorithms that use fixed-size state and continuous action spaces. These algorithms rely on state information similar to the GNN approach but represent it as long vectors with zeros for features corresponding to unconnected EVs. For each combination of training algorithm and experiment scale, at least five training runs were conducted with different random seeds and similar hyperparameters. SAC and TD3 emerged as the best-performing algorithms and were selected for comprehensive testing and validation with both the proposed end-to-end GNN RL architecture (EV-GNN) and a comparative middle-ground approach (FX-GNN). FX-GNN employs the graph formulation solely as a feature extractor to provide high-quality features for subsequent MLP layers, demonstrating that merely using GNNs as feature extractors is better than classic RL algorithms but is insufficient for high-quality solutions Table 1.

### Enabling scalability in RL algorithms

Figure 2 illustrates that classic RL algorithms consistently achieve lower rewards compared to their FX-GNN and EV-GNN counterparts. Specifically, SAC EV-GNN improves the reward by approximately 10.000 units over the classic SAC in the 25 CPs case and more than doubles it in larger-scale scenarios. Similar performance gains are observed when comparing TD3 with TD3 EV-GNN. These results clearly demonstrate that EV-GNN enhances scalability across all cases. Additionally, the FX-GNN feature extractor method outperforms standard SAC and TD3, emphasizing the importance of graph-based state representation and the end-to-end GNN architecture for scalability in the PST problem. For more results, see Supplementary Section 2.1.

### Performance comparison for a large-scale experiment

Evaluating performance metrics beyond reward is crucial in optimal EV charging. Accurately following the power setpoint is essential to avoid grid instability, but user satisfaction is also key. In our problem set-up, user satisfaction measures how closely the SoC of an EV at departure matches the desired (See the definition in Methods Section Eqs. (25–27)). Our collaboration with Dutch CPOs showed us that current methods can not efficiently handle more than a few hundred EVs, despite needing to manage thousands simultaneously. Table 2 shows the results for the 1000 CP case after 100 evaluation episodes, comparing heuristic algorithms, charge As Fast As Possible (AFAP) and Round Robin (RR), with RL models and their FX-GNN and EV-GNN variations. More details on the implementation of AFAP and RR are in Supplementary Section 1.

**Table 2 | Performance comparison of 100 evaluation runs for the complex large-scale 1000 CPs case**

| Algorithm | Energy Charged (MWh) | User Sat. (%) | Energy Error (MWh) |
|---|---|---|---|
| AFAP | 49.8 ± 0.7 | 99.9 ± 0.0 | 22 ± 0.5 |
| RR | 21.7 ± 0.4 | 90.7 ± 0.3 | 33 ± 0.5 |
| A2C | 23.3 ± 0.5 | 87.9 ± 0.4 | 32 ± 0.6 |
| DDPG | 23.4 ± 0.5 | 87.9 ± 0.4 | 32 ± 0.6 |
| TRPO | 1.3 ± 0.0 | 78.4 ± 0.5 | 54 ± 0.8 |
| TQC | 28.9 ± 0.5 | 91.1 ± 0.3 | 27 ± 0.5 |
| SAC | 39.5 ± 0.6 | 95.9 ± 0.3 | 17 ± 0.7 |
| TD3 | 42.7 ± 0.6 | 97.1 ± 0.2 | 17 ± 0.6 |
| SAC FX-GNN | 45.5 ± 0.6 | 98.5 ± 0.1 | 13 ± 0.5 |
| TD3 FX-GNN | 43.3 ± 0.7 | 97.7 ± 0.2 | 12 ± 0.4 |
| SAC EV-GNN | 44.9 ± 0.5 | 98.7 ± 0.1 | 11 ± 0.3 |
| TD3 EV-GNN | 42.3 ± 0.6 | 98.0 ± 0.1 | 13 ± 0.4 |

Existing systems used by CPOs often rely on heuristic methods, which face scalability challenges. For instance, the RR method achieves only 90.7% user satisfaction and results in a 33 MWh energy error, underscoring its limitations in large-scale applications. Classic RL models (A2C, DDPG, TQC, SAC, TD3) show high energy errors, ranging from 17 to 54 MWh, with TRPO performing the worst. FX-GNN improves performance, reducing SAC's and TD3's energy error down to 13 and 12 MWh, and boosting user satisfaction to 98.5% and 97.7%, respectively. The best results come from EV-GNN, which achieves user satisfaction of 98.7% (SAC EV-GNN) and 98% (TD3 EV-GNN), while reducing the energy error to 11 and 13 MWh, respectively. As shown in Table 2, decreasing the PST error indirectly enhances user satisfaction. Therefore, user satisfaction was not explicitly included in the reward function in Equation Eq. (10). These results highlight that EV-GNN significantly enhances the scalability and effectiveness of RL algorithms for large-scale optimization problems such as optimal EV charging with PST.

### Explaining sample efficiency

The overall performance comparison shows that EV-GNN consistently achieves significantly higher rewards, enabling RL algorithms to deliver high-quality solutions for the PST problem in real-time. To understand why EV-GNN performs so well, Fig. 3 provides an explainability analysis from two different perspectives. Figure 3a compares the algorithm outputs for the TD3 algorithm and the TD3 EV-GNN when presented with the same state. Notably, the classic TD3 algorithm generates actions even for CPs without any connected EVs, such as CP 3, 4, 6, and 7, which can lead to invalid actions. In contrast, TD3 EV-GNN inherently avoids generating actions for unoccupied CPs, thus enhancing sample efficiency during training. Moreover, TD3 EV-GNN provides more diverse and personalized actions, ranging from 3 to 10 kW, compared to the classic TD3's less flexible approach. Figure 3b offers a broader view of normalized charging power (action) distributions across different SAC approaches (classic, FX-GNN, EV-GNN) and varying numbers of occupied CPs for the 25 CP PST problem. The classic SAC algorithm exhibits highly polarized action patterns (higher density around 0 and 1), while the FX-GNN approach modifies the action distribution for SAC. Most importantly, EV-GNN introduces substantial changes in SAC's action behavior. This results in a higher chance of selected actions (between 0.2 and 0.8), enhancing control precision and overall performance in the optimal EV charging problem. This effect is further illustrated by the probability $P(0.2 \leq x \leq 0.8)$, which ranges from 13% for classic SAC to 46% for SAC EV-GNN, when more than 67% of the charging stations are occupied, highlighting the model's adaptability to high-demand scenarios.

### Generalization to unseen environments

Deploying RL models in real-world settings frequently necessitates retraining, as differences in state transition probabilities $\mathcal{P}$ between simulators and real environments—or shifts in input distributions, such as changes in EV behavior—can impact performance. Therefore, evaluating the generalization of end-to-end RL models in environments with different state transition probabilities is essential. Figure 4a shows the marginal and joint probability distributions of key variables such as time of stay, arrival time, state of charge at arrival, and departure time. Three new evaluation environments—*small*, *medium*, and *extreme* variations were created to simulate different degrees of deviation from the training environment, such as applying an RL model trained in one city to another with different charging patterns.

The generalization capabilities of AFAP, SAC, TD3, and their FX-GNN and EV-GNN variations are evaluated in the 500 CP scenario (Fig. 4b). As expected, greater deviations in the environment lead to lower episode rewards for all methods. However, EV-GNN consistently outperforms the others, showing superior generalization. For instance, in extreme environments, TD3 EV-GNN achieves more than two times better reward than classic TD3. In contrast, SAC FX-GNN and TD3 FX-GNN show worse or similar generalization compared to their classic versions, likely overfitting to the training environment. This pattern holds across all environments, highlighting that while FX-GNN improves feature extraction, it does not address the broader challenges of scaling and adaptation that the full EV-GNN approach effectively manages.

### EV-GNN for multi-discrete problems

The optimal EV charging problem with PST involves controlling the charging current for CPs, ranging from zero to a maximum value, typically modeled as a continuous action space $\boldsymbol{a} \in [0, 1]$. However, practical constraints often limit this to discrete current levels based on charging station technology[40], such as a Type-2 charger with levels $I^{ch} \in \{0, 6, 8, \ldots, 32\}$. When only discrete actions are possible, the PST problem is represented by a multi-discrete action space $\mathcal{A} = \mathcal{A}_1 \times \cdots \times \mathcal{A}_J$, where $\mathcal{A}_j = \{0, \underline{I}^{ch}, \overline{I}^{ch}\}$ defines the discrete set for each CP $j \in J$. A2C, TRPO, PPO, Mask PPO[41], and recurrent PPO[42] are RL algorithms designed for multi-discrete action spaces, whereas TD3 was not initially intended for such problems. However, by incorporating EV-GNN, TD3 can be adapted for use in multi-discrete and discrete action spaces. In experiments comparing TD3 EV-GNN with classic multi-discrete RL methods, Fig. 5 shows TD3 EV-GNN significantly outperforms the baselines. For 25 CPs, TD3 EV-GNN achieves around three times higher reward than the second best (TRPO), and for 100 CPs, the increase is even higher. This highlights the scalability and superior performance of TD3 EV-GNN in multi-discrete scenarios.

### Application to V2G profit maximization

Up to now, our experiments have focused solely on solving the PST problem. However, a CPO can face problems with multiple objectives as charging technology advances. Currently, most EVs rely exclusively on the power grid for charging. However, with the anticipated increase in bidirectional EVs and chargers capable of vehicle-to-grid (V2G) interactions, the landscape will change. In the near future, EVs will not only draw energy from the grid but also return it, constituting invaluable flexible loads to support the operation of the grid while earning compensation for these services[43]. This evolution means that CPOs will need to optimize EV charging schedules while also considering factors such as energy costs and grid loads.

In this section, we show that the EV-GNN approach extends beyond the PST problem and can efficiently address more complex EV charging challenges, such as V2G profit maximization. This problem involves managing residential loads, PV contributions, and demand response events. Two sets of experiments were conducted to evaluate the performance of various approaches: one on a smaller scale with 25 CPs and another on a much larger scale with 500 CPs. Figures 6a, b illustrate the training performance of these methods, focusing on maximum and average rewards. In

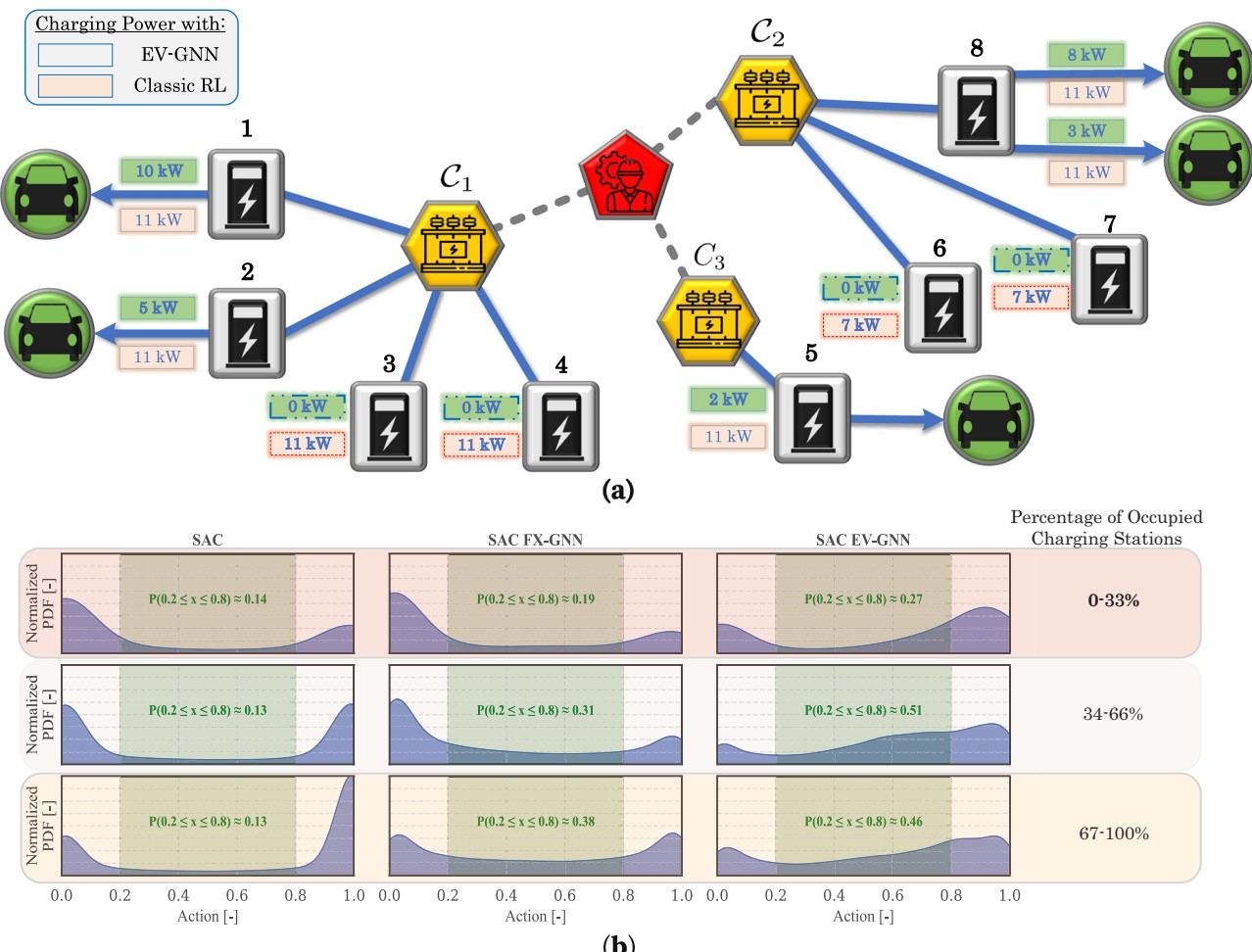

**Fig. 3 | Explainability analysis for EV-GNN in the 25 CP case. a** Comparison of the actions generated by classic RL algorithm (TD3) and its EV-GNN extension (TD3 EV-GNN). Through graph pruning, EV-GNN ensures that only valid actions are produced, avoiding impossible actions such as charging when no EV is connected. In contrast, the classic TD3 generates actions that may not be feasible, e.g., classic TD3 assigns charging power to charging stations 3, 4, 6, and 7 even when there is no EV connected. **b** Visualization of the normalized action probability density functions for three distinct state groups, categorized by EV occupancy levels at charging stations (up to 33%, 66%, and 100%). This shows how the EV-GNN approach enables RL algorithms to generate more diverse and state-specific actions, highlighted by the probability $P(0.2 \leq x \leq 0.8)$ of an action falling within the [0.2, 0.8] range. This unique capability of EV-GNN enhances the model's effectiveness in achieving learning objectives.

line with the findings from the PST problem, both TD3 EV-GNN and SAC EV-GNN significantly outperform their classic counterparts in terms of maximum reward. The FX-GNN methods also demonstrate superior performance compared to classic SAC and TD3 in the 25 CP scenario. Additionally, Fig. 6c presents the average performance results from 100 evaluation runs for the 500 CP case. This comparison includes two heuristic algorithms (AFAP and RR), two mathematical programming methods (MPC and Optimal), as well as the RL methods. This comprehensive analysis highlights the relative strengths and weaknesses of each approach in handling large-scale CP management.

As anticipated, the AFAP and RR methods yield lower profits since they do not utilize discharging capabilities, resulting in approximately 5 MWh of overloads. Despite this, they achieve 100% user satisfaction. In contrast, the EV-GNN methods excel in maximizing profits with minimal overloads, though this comes at the expense of user satisfaction. FX-GNN and classic RL methods generate about half the profits compared to EV-GNN approaches, but offer roughly 10% higher user satisfaction. Among the non-heuristic methods, MPC performs second only to EV-GNN in terms of profits and provides the highest user satisfaction. However, MPC struggles with transformer overload constraints due to its extensive search space and time limitations. This complexity is underscored by the problem's scale: 500 CPs, 35 transformers, and demand response events that last 1 hour

per day and can reduce available transformer capacities by up to 15%. Notably, MPC requires approximately 5 minutes per step to generate a charging schedule, whereas RL methods can produce results in less than a second. This significant time advantage highlights the importance of RL methods for real-time optimal EV charging.

## Discussion

Our results highlight EV-GNN as an effective solution for large-scale EV charging optimization, addressing the limitations of traditional methods. While mathematical programming approaches are computationally intensive and classic RL methods fail with complex, high-dimensional dynamics, EV-GNN facilitates real-time decision-making by leveraging a GNN-based architecture suited for large state-action spaces. For CPOs managing thousands of EVs daily, balancing grid stability and user SoC requirements, EV-GNN represents a practical advancement, enabling scalable, high-quality solutions that outperform conventional approaches in both efficiency and applicability.

Our experimental results show that EV-GNN scales more effectively and generalizes better to unseen environments compared to traditional RL algorithms, a crucial ability for real-world applications where systems must adapt to changing conditions. EV-GNN's improved scalability and generalization stem from its end-to-end GNN architecture, which efficiently

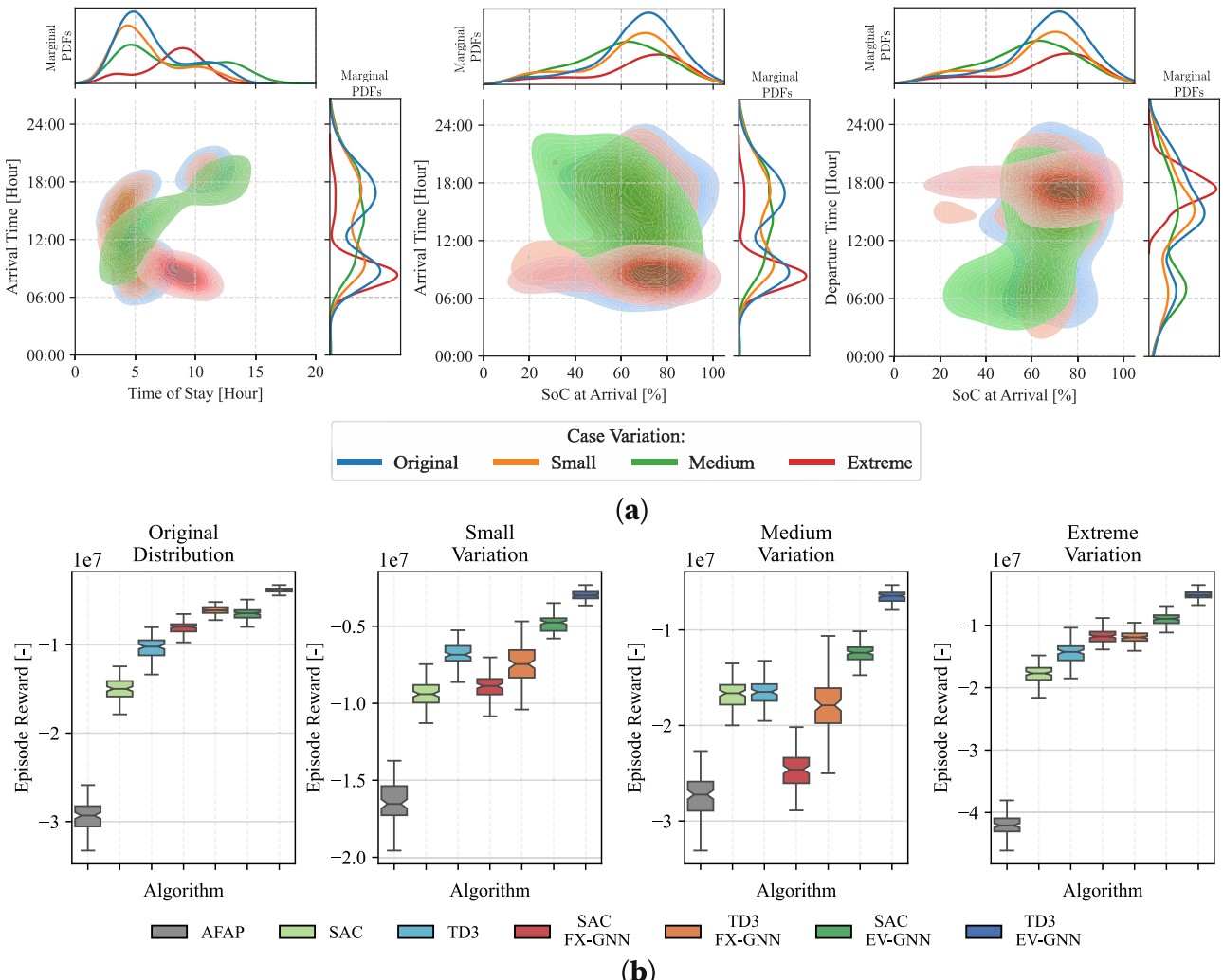

**Fig. 4 | Generalization analysis for out-of-training-distribution environments.**
**a** Joint and marginal probability density functions of key environmental variables affecting state transition probability $\mathcal{P}$, including EV time of stay, time of arrival, SoC at arrival, and departure time. The cases depicted are: *Original* (the environment used for training), *Small*, *Medium*, and *Extreme*, which represent increasing degrees of variation from the original environment. **b** To assess the generalization capabilities of the proposed methods, all algorithms were trained on the environment using the original distribution and then evaluated on its modified versions. The box plots show the mean and standard deviation of the achieved rewards after 100 evaluation runs across four different environments for the 500 CP case.

filters irrelevant information, such as zero values for absent EVs, enhancing sample efficiency during training. Additionally, the GCN architecture leverages graph symmetries, allowing it to generalize across similar structures by focusing on node features and connectivity, regardless of spatial arrangement. For instance, a branch with a transformer connected to three chargers, each with one EV, is symmetrical to other branches with the same configuration but potentially differing locations or arrangements. This allows the GCN to generalize effectively across similar structures within the graph.

This paper demonstrates that the EV-GNN methodology excels in both continuous and multi-discrete settings, and is also adaptable to discrete problems. The successful integration of TD3 with EV-GNN highlights its scalability and effectiveness in handling complex multi-discrete domains. Importantly, the EV-GNN approach is highly versatile, making it compatible with a wide range of deep RL algorithms, whether they involve continuous or discrete control. Notably, EV-GNN can achieve higher rewards in the challenging V2G profit maximization scenario, which incorporates loads, PV, and demand response events. This performance underscores the methodology's robustness and ability to tackle complex real-world problems efficiently.

The significant improvements in performance, scalability, and generalization shown in this study open promising avenues for future research in optimal EV charging and beyond. EV-GNN's success in both PST and

V2G profit maximization problems suggests further evaluation is needed in other EV charging scenarios. Future work could also apply EV-GNN's dynamic graph and end-to-end GNN principles to resource allocation problems with dynamic state spaces, such as vehicle routing[44], portfolio optimization, and production planning[45]. Additionally, enhancing EV-GNN to ensure constraint satisfaction, potentially by integrating mathematical programming[46] or Safe RL strategies[47], could improve its practical applicability for real-world optimization challenges.

## Methods
### Graph MDP for optimal EV charging
The centralized optimal EV charging problem, from the perspective of the CPO agent using PST, is formulated as an MDP within the graph domain and is represented as: $\mathcal{M} = (\mathcal{S}, \mathcal{A}, \mathcal{P}, \mathcal{R})$, where $\mathcal{S}$ is the state space, $\mathcal{A}$ the action space, $\mathcal{P}$ the transition probability, and $\mathcal{R}$ the reward function. The state space $\mathcal{S}$ is defined as the graph $\mathcal{G} = (N, \mathcal{E})$ at step $t$, consisting of observations for each type of node in the problem. Each EV node, representing an EV connected to CP $j$, charger $i$, and transformer $w$, has a feature vector:

$$\boldsymbol{x}^{\text{ev}} = [z, E - E^{\text{arr}}, t - t^{\text{arr}}, j, i, w], \tag{5}$$

where $z$ is a binary variable indicating whether the EV is fully charged, $E - E^{arr}$ measures the energy transferred to the EV since connection, and $t - t^{arr}$ measures the time elapsed since the connection of the EV. The variables $j, i, w$ serve as unique identifiers (port, charger, transformer group) to distinguish each EV node from others of the same type. Note that in the realistic public PST problem, the CPO is unaware of the SoC and the EV's arrival and departure times, as the communication protocol between the EV and the charger does not share this information. Each charging node,

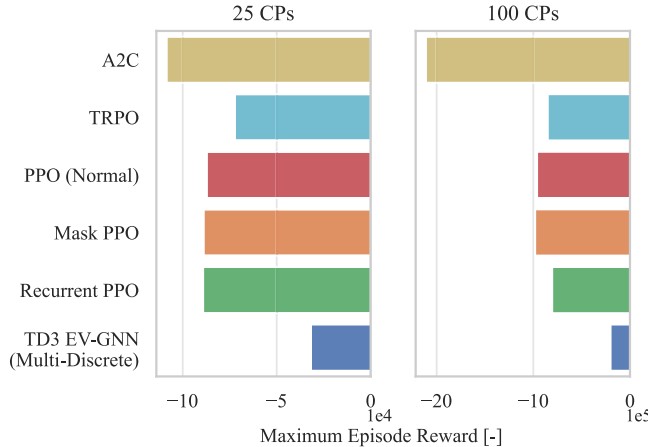

representing a charger $i$, is characterized by the following feature vector:

$$\boldsymbol{x}^{cs} = [\underline{I}, \overline{I}, |J|, i], \tag{6}$$

with $\underline{I}$ and $\overline{I}$ denoting the minimum and maximum charging currents that the charger can handle, and $|J|$ indicates the number of CPs available on the charger. These features are crucial as they define the charger's operational limits and capacity. A simpler feature vector represents transformer nodes:

$$\boldsymbol{x}^{tr} = [\overline{P}, w], \tag{7}$$

where $\overline{P}$ is the maximum power capacity of the transformer, and $w$ is the unique transformer identifier. This information is essential for controlling the transformer's power usage, ensuring it does not exceed its capacity. Finally, the CPO node is described by a more comprehensive feature vector:

$$\boldsymbol{x}^{cpo} = \left[ \frac{d}{7}, \sin\left(\frac{h}{48 \cdot \pi}\right), \cos\left(\frac{h}{48 \cdot \pi}\right), P_t^{set}, P_{t-1}^{tot} \right], \tag{8}$$

The day $d$ and hour $h$ encoding allows the model to learn daily and hourly variations in energy demand and EV availability. $P_t^{set}$ indicates the power setpoint for the current time step, while $P_{t-1}^{tot}$ reflects the actual power usage from the previous time step, acting as important feedback for adjusting future power allocations, thereby enhancing the efficiency of power management.

The CPO agent's action space $\mathcal{A}$ is represented as a dynamic vector with a size equal to the number of EV nodes in the graph $\mathcal{G}$. Each element of this vector corresponds to the charging current allocated from CP $j$ to its connected EV. Specifically, for each EV, the action $a_j$ is a continuous variable taking values in the interval $[0, 1]$. The actual charging current $I_j$ supplied by

**Fig. 5 | Maximum episode reward obtained for classic multi-discrete RL algorithms and TD3 EV-GNN.** The maximum episode reward is the best reward achieved by each algorithm after 5 training runs with different random seeds. For the TD3 EV-GNN approach, the last GCN layer is configured with three features per node ($F_L = 3$), corresponding to the number of discrete actions allowed.

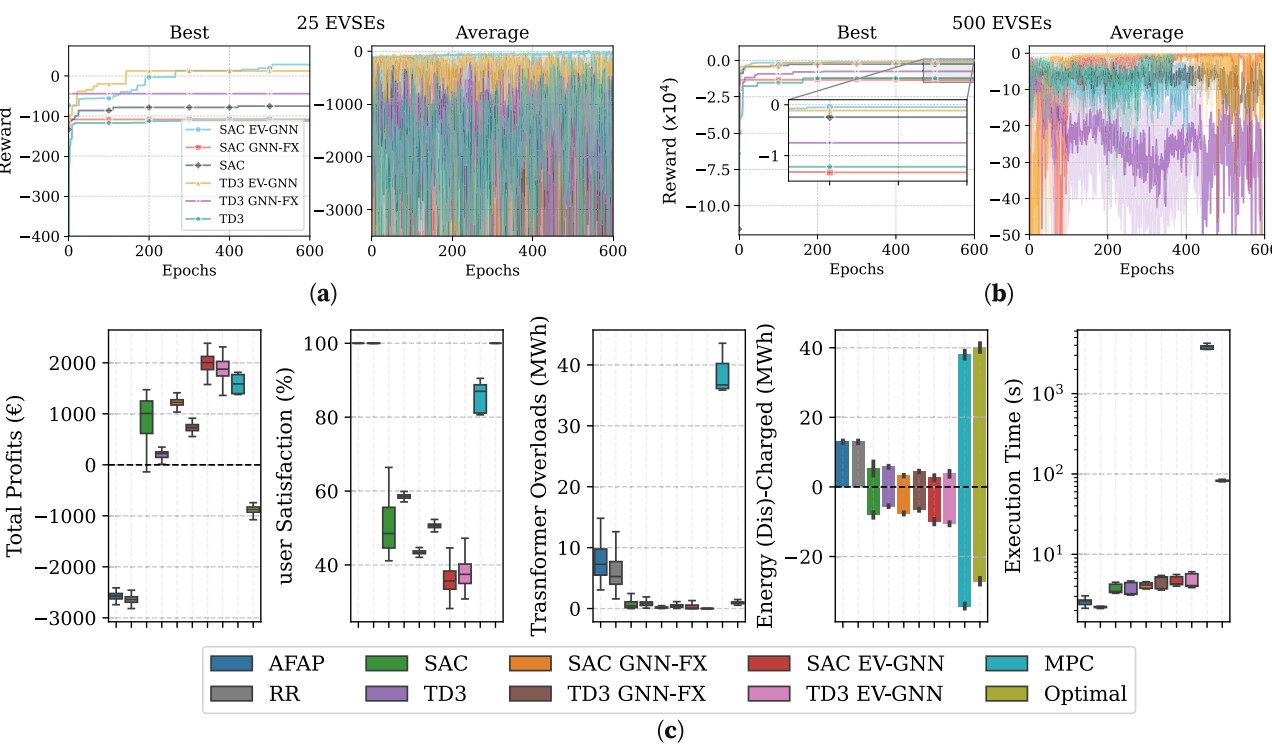

**Fig. 6 | V2G profit maximization with loads, PV, and demand response events. a** Training performance of baseline and enhanced RL algorithms for 25 CPs, showing the best and average rewards achieved by various methods, including GNN-enhanced approaches. **b** Training Performance in a large-scale case study with 500 CPs. **c** Performance comparison for 100 simulations with 500 CPs in the V2G Profit Maximization problem. This panel compares various approaches, including heuristic methods (AFAP and RR), baseline RL algorithms, GNN-enhanced RL methods, an MPC method, and an optimal solution assuming future knowledge. Key metrics include total profit, user satisfaction (reflecting how closely the EV's SoC at departure meets the desired level), transformer overloads, and execution time, which indicates how long each algorithm takes to produce an optimization solution. The boxes show the quartiles of the results while the whiskers extend to show the rest of the distribution, except for points that are determined to be "outliers".

the CP to the EV is given by:

$$I_j = a_j \cdot \overline{I}, \tag{9}$$

where $\overline{I}$ is the maximum allowable current that the CP can provide. For instance, if $a_j = 0$, it indicates that EV will not charge at that time. Conversely, if $a_j = 1$, it implies that the EV is receiving the maximum charging current available from the CP. This continuous range of action values allows for flexible and precise control over the charging process.

The intricate dynamics of the coordinated EV charging process are captured by the state transition probability function, $\mathcal{P}$. Although it is often unknown and challenging to model $\mathcal{P}$ explicitly, the RL agent learns it through continuous interaction with the environment. In detail, $\mathcal{P}$ models the uncertainty and variability inherent in the system, including factors such as the arrival and departure times of EVs, the specific characteristics of each EV, and fluctuations in charging prices based on the wholesale energy market.

The reward function $\mathcal{R}$ plays a vital role in guiding the learning process of RL algorithms, and it directly aligns with the objective of the optimization problem as it is designed to drive the system toward desired outcomes. In the PST problem, the reward function is defined as:

$$R(s, a) = -\left(P_{t-1}^{\text{set}} - P_{t-1}^{\text{tot}}\right)^2. \tag{10}$$

The reward function evaluates the actions' performance by penalizing the squared error between the setpoint and the actual power usage. A lower squared error results in a higher reward, encouraging the system to more accurately match the setpoint to the actual usage, which is crucial for effective PST. In our scenario, minimizing the tracking error naturally aligns with maximizing the SoC for all controlled EVs, provided the power setpoint is sufficiently large. As a result, the reward function does not require an explicit term to maximize user satisfaction.

## Simulation environment

Accurate experimentation with optimal EV charging and RL necessitates the use of a simulator based on real-world data. This ensures that the results obtained are relevant and reliable for assessing the performance of the algorithms, thereby providing meaningful insights into their practical applicability and effectiveness. For this reason, the EV2Gym simulator environment[48] was used to model the EV charging problems and run the RL simulations. EV2Gym leverages real EV charging transactions from ElaadNL[49], incorporates real EV characteristics[50], and uses actual electricity prices[51], ensuring that the simulations closely reflect real-world conditions.

Each simulation run for the PST problem consisted of 115 steps, each lasting 15 minutes, starting at 05:00 in the morning. Each run commenced on a different day of the year, resulting in variations in prices, power setpoints, EV behavior, and EV characteristics. The power setpoint was generated at the start of the simulation to ensure enough energy for all EVs to charge by shifting their loads over time in a price-based manner. This generation process was randomized for each run, leading to distinct power setpoints and numbers of EVs every time. The experiments were structured by dividing the CPs into transformer groups: three groups for the 25 CP case, seven groups for the 100 CP case, thirty-five groups for the 500 CP case, and seventy groups for the 1000 CP case.

## Training and parameterization

The training of RL models continued until learning plateaued. In practice, the training process consisted of around 20,000 epochs, each with 112 steps. After every 300 runs, 100 evaluation runs were conducted to assess performance. The RL models were frequently evaluated during training, and the best-performing model from these evaluation rounds was selected.

Training epochs were empirically validated through repeated evaluations, confirming that additional epochs did not yield improved performance.

RL training was performed on the Delft Blue HPC system[52], which enabled large-scale simulations with up to 1000 EV chargers and allowed us to explore substantially larger state-action spaces. For our most intensive experiments, we required around 120 GB of RAM to store the RL algorithms' replay buffer, 2 CPU cores, and a single GPU. Importantly, these simulations can also be run on smaller systems by implementing more efficient memory management or batch processing techniques, such as offloading portions of the replay buffer to a hard drive. This approach optimizes resource utilization and makes high-fidelity simulations accessible even without a full-scale supercomputer, albeit at the cost of additional computational time depending on the specific application.

The experiments used the StableBaselines3 implementation of the classic RL algorithms[53], while the code for the FX-GNN and EV-GNN is provided. Python 3.11, PyTorch 2.3.1, and PyTorch Geometric 2.5.3 were used for training the NNs and GNNs, while Gurobi 11.0.2 was used to derive the solutions for the MPC and the Optimal algorithms.

The classic RL algorithms, the FX-GNN, and EV-GNN variations used the same default hyperparameters for learning rate, batch size, etc. The classic RL algorithms employed actor and critic NNs with two hidden layers of 512 nodes each. Various configurations with different numbers of layers and hidden nodes were tested, but these changes did not result in better learning performances. SAC FX-GNN and TD3 FX-GNN used independent GNN feature extractors with $F_0 = 32$ for the 25 and 100 CP cases and $F_0 = 64$ for the 500 and 1000 CP cases. These were followed by three GCN layers with feature dimensions $F_l \in \{64, 128, 256\}$ and two fully connected layers with 512 nodes each. The EV-GNN algorithms adopted the same feature extractor architecture ($F_0$) as FX-GNN but included $L$ GCN layers for the actor and $K$ GCN layers for the critic. For the 25 and 100 CP cases, $K = L = 3$, with actor feature dimensions $F_l \in \{64, 32, 1\}$ and critic feature dimensions $F_k \in \{64, 128, 192\}$. In the 500 CP case, $K = L = 3$ with $F_0 = 64$, the actor's GCN layers had feature dimensions $F_l \in \{128, 64, 1\}$, and the critic's dimensions were $F_k \in \{128, 256, 384\}$. In the 1000 CP case, optimal performance was achieved with $K = 6$ and $L = 5$, with $F_0 = 64$. The actor's GCN feature dimensions $F_l \in \{128, 256, 384, 256, 128, 1\}$, and the critic's dimensions $F_k \in \{128, 256, 384, 512, 640\}$.

In all cases, the critic network in EV-GNN was followed by two fully connected layers with 512 nodes each. End-to-end GCN architectures are significantly more lightweight, resulting in significantly fewer trainable parameters compared to full MLP architectures. For instance, in the 1000 CP case, the actor NN of SAC EV-GNN had only 256, 514 trainable parameters, whereas the classic SAC had 2, 826, 704.

## MIP formulation of optimal EV charging

PST is the problem where CPOs manage multiple chargers, either procuring energy in advance or operating under limited capacity contracts, and strive to adhere to the power setpoint for efficient and fair energy distribution. While the arrival and departure times, as well as the SoC are unknown, it is assumed that an EV is fully charged when no energy exchange is measured through the CP. Therefore, the PST problem can be formulated as an MIP problem described by Eqs. (11–24), for all $w \in \mathcal{W}, j \in \mathcal{J}, i \in \mathcal{C}$, and $t \in \mathcal{T}$.

$$\min_{I_{j,i,t}^{\text{ch}}, I_{j,i,t}^{\text{dis}}} \sum_{t \in \mathcal{T}} \left(P_t^{\text{set}} - P_t^{\text{tot}}\right)^2 \tag{11}$$

Subject to:

$$P_t^{\text{tot}} = \sum_{i \in \mathcal{C}} \sum_{j \in \mathcal{J}} \left(P_{j,i,t}^{\text{ch}} + P_{j,i,t}^{\text{dis}}\right) \quad \forall j, \forall i, \forall t \tag{12}$$

$$P_{j,i,t}^{\text{ch}} = I_{j,i,t}^{\text{ch}} \cdot V_{j,i,t} \cdot \sqrt{\phi_{j,i,t}} \cdot \eta_{j,i,t}^{\text{ch}} \cdot \omega_{j,i,t}^{\text{ch}} \quad \forall j, \forall i, \forall t \tag{13}$$

$$P_{j,i,t}^{\text{dis}} = I_{j,i,t}^{\text{dis}} \cdot V_{j,i,t} \cdot \sqrt{\phi_{j,i,t}} \cdot \eta_{j,i,t}^{\text{dis}} \cdot \omega_{j,i,t}^{\text{dis}} \quad \forall j, \forall i, \forall t \tag{14}$$

$$\underline{E}_{j,i} \leq E_{j,i,t} \leq \overline{E}_{j,i} \quad \forall j, \forall i, \forall t \tag{15}$$

$$E_{j,i,t} = E_{j,i,t-1} + (P_{j,i,t}^{\text{ch}} + P_{j,i,t}^{\text{dis}}) \cdot \Delta t \quad \forall j, \forall i, \forall t \tag{16}$$

$$E_{j,i,t} = E_{j,i,t}^{\text{arr}} \quad \forall j, \forall i, \forall t \,|\, t = t_{j,i,t}^{\text{arr}} \tag{17}$$

$$\underline{I}_{j,i}^{\text{ch}} \leq I_{j,i,t}^{\text{ch}} \leq \overline{I}_{j,i}^{\text{ch}} \quad \forall j, \forall i, \forall t \tag{18}$$

$$\underline{I}_{j,i}^{\text{dis}} \geq I_{j,i,t}^{\text{dis}} \geq \overline{I}_{j,i}^{\text{dis}} \quad \forall j, \forall i, \forall t \tag{19}$$

$$I_{i,t}^{\text{cs}} = \sum_{j \in \mathcal{J}} \left( I_{j,i,t}^{\text{ch}} \cdot \omega_{j,i,t}^{\text{ch}} + I_{j,i,t}^{\text{dis}} \cdot \omega_{j,i,t}^{\text{dis}} \right) \quad \forall j, \forall i, \forall t \tag{20}$$

$$\underline{I}_i^{\text{cs}} \leq I_{i,t}^{\text{cs}} \leq \overline{I}_i^{\text{cs}} \quad \forall i, \forall t \tag{21}$$

$$P_{w,t}^{\text{EVs}} = \sum_{i \in \mathcal{C}_w} \sum_{j \in \mathcal{J}} \left( P_{j,i,t}^{\text{ch}} + P_{j,i,t}^{\text{dis}} \right) \quad \forall w, \forall j, \forall i, \forall t \tag{22}$$

$$\underline{P}_{w,t}^{\text{tr}} \leq P_{w,t}^{\text{EVs}} + P_{w,t}^{\text{L}} + P_{w,t}^{\text{PV}} \leq \overline{P}_{w,t}^{\text{tr}} - P_{w,t}^{\text{DR}} \quad \forall w, \forall t \tag{23}$$

$$\omega_{j,i,t}^{\text{ch}} + \omega_{j,i,t}^{\text{dis}} \leq 1 \quad \forall j, \forall i, \forall t \tag{24}$$

This formulation aims to minimize the squared power tracking error, defined as the squared difference between the procured or setpoint power $P_t^{\text{set}}$ and the actual power $P_t^{\text{tot}}$ at time $t$. The current of a single CP $j$ is modeled using two decision variables, $I^{\text{ch}} \cdot \omega^{\text{ch}}$ and $I^{\text{dis}} \cdot \omega^{\text{dis}}$, where $\omega^{\text{ch}}$ and $\omega^{\text{dis}}$ are binary variables, to differentiate between charging and discharging behaviors. The charging current and power ($I^{\text{ch}}$ and $P^{\text{ch}}$) are positive, while the discharging current and power ($I^{\text{dis}}$ and $P^{\text{dis}}$) are negative. Equations (13) and (14) define power relationships of currents with CP phase voltage $V$, number of phases $\phi$, and (dis)-charging efficiency $\eta$, while (15–17) address EV battery constraints during operation with a minimum and maximum capacity of $\underline{E}, \overline{E}$, and energy $E^{\text{arr}}$ at time of arrival $t^{\text{arr}}$. Equations (18) and (19) impose current charging and discharging limits (with minimum current $\underline{I}$ and maximum $\overline{I}$ for each EV and CP, with (21) applying to the entire charger. The transformer power constraint is specified in (23) as a function of total EV load $P^{\text{EV}}$, inflexible loads $P^{\text{L}}$, PV generation $P^{\text{pv}}$ and curtailed power due to demand response (DR) events $P^{\text{DR}}$. To prevent simultaneous charging and discharging, the binary variables $\omega^{\text{ch}}$ and $\omega^{\text{dis}}$ are constrained by (24).

Furthermore, the following evaluation metrics are used in this study. The user satisfaction metric measures how closely the SoC of an EV at departure matches its target SoC*. For a set of EVs $\mathcal{M}$, user satisfaction is given by:

$$\text{User Sat.} = \frac{1}{|\mathcal{M}|} \cdot \sum_{m \in \mathcal{M}} \left( \frac{\text{SoC}_m}{\text{SoC}_m^*} \right) \cdot 100\%. \tag{25}$$

Energy error measures the discrepancy between the procured power and the actual power used for charging, and is defined as:

$$\text{Energy Error} = \sum_{t \in \mathcal{T}} |P_t^{\text{set}} - P_t^{\text{tot}}| \cdot \Delta t, \tag{26}$$

where $\Delta t$ is the duration of each time step. Additionally, the total energy charged represents the total amount of energy supplied to EVs over time and

is defined as:

$$\text{Total Energy Charged} = \sum_{t \in \mathcal{T}} P_t^{\text{tot}} \cdot \Delta t \tag{27}$$

## MIP and MDP for the V2G profit maximization problem

V2G profit maximization is the second problem explored, which aims to maximize the profits of a CPO while fully meeting the demands of EV users. Unlike the PST problem, this scenario assumes that when an EV arrives at charging station $i$ and CP $j$, it communicates its departure time $t_{j,i}^{\text{dep}}$ and desired battery capacity at departure $E_{j,i}^*$. Additionally, the battery capacity $E_{j,i,t}$ of each EV is known while it is connected to the charger. These assumptions are typically made in research because this information can be obtained from EVs as more advanced communication protocols are developed. The objective function, detailed in (28), depends on the charging ($c^{\text{ch}}$) and discharging prices ($c^{\text{dis}}$) for each CP $j$, $i$.

$$\max_{I_{j,i,t}^{\text{ch}}, I_{j,i,t}^{\text{dis}}} \sum_{t \in \mathcal{T}} \sum_{i \in \mathcal{C}} \left( -P_{i,t}^{\text{ch}} \cdot c_{i,t}^{\text{ch}} + P_{i,t}^{\text{dis}} \cdot c_{i,t}^{\text{dis}} \right) \cdot \Delta t \tag{28}$$

Subject to constraints from Eq. 15–28 (main text) and:

$$E_{j,i,t} \geq E_{j,i,t}^* \quad \forall j, \forall i, \forall t \,|\, t = t_{j,i,t}^{\text{dep}} \tag{29}$$

V2G profit maximization with loads is more complex than the PST problem due to the added uncertainties of PV generation and demand response events. These events are communicated only half an hour before they occur, making it challenging to comply with all the problem constraints.

With V2G technology enabling precise communication, CPOs will always have up-to-date information on the SoC of each EV and their departure times[54]. Consequently, the state information for the EV feature vector in this problem changes and is defined as follows:

$$\boldsymbol{x}^{\text{ev}} = [\text{SoC}_t, t^{\text{dep}} - t, j, i, w], \tag{30}$$

where $t^{\text{dep}} - t$ is the steps remaining until the EV departs, and $j$, $i$, $w$ are unique identifiers referring to the CP, charger, and transformer group the EV belongs. The charging station and transformer feature vectors, $\boldsymbol{x}^{\text{cs}}$ and $\boldsymbol{x}^{\text{tr}}$, are the same as the PST problem, as shown in Eqs. (6, 7), while the CPO node features are defined as:

$$\boldsymbol{x}^{\text{cpo}} = \left[ \frac{d}{7}, \sin\left( \frac{h}{48 \cdot \pi} \right), \cos\left( \frac{h}{48 \cdot \pi} \right), c_t^{\text{ch}}, P_{t-1}^{\text{tot}} \right], \tag{31}$$

with $c_t^{\text{ch}}$ being the electricity price for charging at step $t$, while the price for discharging is a linear combination of the charging price, in our experiments $c_t^{\text{dis}} = -1.2 \cdot c_t^{\text{ch}}$. In the V2G profit maximization problem, discharging is possible. Hence, the action space takes values in $[-1, 1]$ with negative values signifying discharging and positive charging relative to the maximum CP capacity. Finally, the reward function is based on existing literature on V2G profit maximization[48] and is defined $\forall w, \forall j, \forall i$ as:

$$R(s, a) = c_t - 100 \cdot \epsilon_t^{\text{ov}} - 100 \cdot \exp\left( -10 \cdot \epsilon_t^{\text{usr}} \right), \tag{32}$$

where $c_t$ are the total costs and profits obtained by charging and discharging, $\epsilon_t^{\text{ov}}$ is the amount of power exceeding the limit of power transformer overload during the passed step in kW, and $\epsilon_t^{\text{usr}}$ represents the user satisfaction score of EVs departing at the current step, defined as the ratio of the current SoC to the desired SoC*, indicating how closely the

**Article**

EVs' charge levels match their target. This reward function motivates the RL agent to maximize profits and user satisfaction while heavily penalizing overloadings.

## Data availability

All experiments were conducted using the EV2Gym simulator[48], and the results are reproducible via the configurations provided at: https://github.com/StavrosOrf/EV-GNN.

## Code availability

Access the open-source code, along with trained models, at https://github.com/StavrosOrf/EV-GNN and https://github.com/distributionnetworksTUDelft/EV-GNN.

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

## Acknowledgements
This work is funded by the HORIZON Europe Drive2X Project 101056934. Also, the work is partially funded by the TESTBED-2/EU MSCA 872172.

## Author contributions
S.O. and P.P.V. were responsible for the methodology, conceptualization, visualization, drafting the initial manuscript, and revising the draft. V.R. and E.M.S. contributed to visualization and manuscript review. P.P.V. and P.P. supported the acquisition of funding.

## Competing interests
The authors declare no competing interests.
