## [Transparent Peer Review file · Communications Engineering]

Scalable Reinforcement Learning for Large-Scale Coordination of Electric Vehicles Using Graph Neural Networks

Corresponding Author: Mr Stavros Orfanoudakis

Version 0:

Reviewer comments:

Reviewer #1

(Remarks to the Author)

The paper introduces EV-GNN, a novel approach leveraging Graph Neural Networks (GNN) and reinforcement learning (RL) to optimize large-scale electric vehicle (EV) charging management. The main goal is to address scalability and uncertainty challenges posed by increasing EV adoption, enhancing energy efficiency.

The paper's major claims are:

- Scalability of EV-GNN: The proposed framework can manage large-scale electric vehicle (EV) charging scenarios effectively by reducing power tracking errors.
- Improved Performance: their results show that EV-GNN outperforms traditional reinforcement learning (RL) methods and heuristic approaches in terms of scalability, energy efficiency, and user satisfaction.
- Innovative Use of GNNs: The combination of Graph Neural Networks (GNNs) and RL offers a novel (and in my honest opinion interesting) way to model the dynamic relationships between vehicles, charging stations, and the electricity grid.

The claims of this paper are novel and relevant in the field of EVs. The integration of GNNs into EV charging management is an innovative application, as this method has rarely been explored in this context as far as I know. Additionally, the work addresses pressing challenges in the EV and energy management community, including scalability and dynamic decision-making in large networks.

Their simulations also integrate real-world data on charging transactions, energy prices, and vehicle characteristics. Their experiments scenarios with small-scale (25 stations) and large-scale (up to 1000 stations) setups were simulated to assess performance and scalability. Simulations are performed on a supercomputer using Nvidia GPUs and high-performance CPUs.

This raises a small problem: simulations require high-performance computing infrastructure, limiting replicability in less-equipped settings. A deeper analysis of computational costs versus the benefits achieved, especially in resource-limited contexts, could add more practical insights.

Suggestion of possible improvement: An exploration of how variations in key parameters (e.g., number of vehicles, charging rates) affect the performance of EV-GNN could provide deeper insights into its robustness.

In conclusion, the paper is well-written, and furthermore I appreciate the diagrams and the result plots (especially Figure 4).

Reviewer #3

(Remarks to the Author)

The authors have addressed a worthy topic and have proposed an innovative solution. However, a few comments from this reviewer will need to be addressed before the manuscript can be considered ready for publication.

1. A clear, concise and specific statement of the study objective seems to be missing. There exist several sentences that allude to this but are rather vague.
2. The EV part of EV-GNN is really about the application context, not the methodology, right? Is this appropriate?
3. Figure 1 caption may be shortened significantly. Material could be placed in the main text.
4. All symbols in the expressions and equations need to be defined.

5. Acronyms must be defined at their first instance of use.
6. Different notations are used to denote the reward function. Need for consistency.
7. Is there any reason why the reward function is a single attribute function? Would that produce a really robust solution?
8. Stronger evidence of the efficacy of the developed method, will be useful to the reader. What is the baseline? What is the integrity of the baseline?

Version 1:

Reviewer comments:

Reviewer #1

(Remarks to the Author)

I believe that the modifications and additions made following the review are quite satisfactory and have indeed improved the paper.

Reviewer #2

(Remarks to the Author)

Thank you very much for your well-written, detailed, and very interesting manuscript. As a computational person myself, dealing with scalability of reinforcement learning is of utmost importance in my area of expertise. In addition to that, the application domain picked is very well-suited to the methodology; something that is sometimes rare, and it adds value to your contribution.

Since I am reviewing a second iteration, the manuscript has been significantly improved as far as its presentation and its readability, so my comments are very limited. This also speaks to how well-written and interesting the manuscript is overall. Thank you for this contribution.

1. In the literature review, the authors have primarily focused on computational studies, omitting some research in transportation science or the nexus of transportation and other infrastructure systems. I would recommend that the authors see some of the recent contributions in scheduling (see, e.g., “Electric vehicle community charging hubs in multi-unit dwellings: Scheduling and techno-economic assessment” by Zheng et al., or “Leveraging electric vehicles as a resiliency solution for residential backup power during outages” by Liu, Vlachokostas, and Kontou).

2. In Equation (3), it appears that the index i takes on values 1 to L ; hence it would need to be written as $i = 1, \dots, L$ rather than with the use of the “belongs to” sign ($i \in 1, \dots, L$).

3. I may have missed that (and I apologize if I did; your results discussion was very comprehensive overall): how do you ensure that your training epochs are enough to reach a steady-state in RL?

Thank you again for your fantastic contribution. I look forward to seeing it finalized.

“Scalable Reinforcement Learning for Large-Scale Coordination of Electric Vehicles Using Graph Neural Networks”

Stavros Orfanoudakis, Valentin Robu, E. Mauricio Salazar, Peter Palensky, and Pedro P. Vergara

Dear Editors and Reviewers,

We greatly appreciate your response and attention to our manuscript submission. The reviewers' insightful feedback has significantly improved our paper's quality. We carefully considered all comments and suggestions during the revision process, resulting in substantial improvements to the article. Below, you can find specific answers to every reviewer's comment, highlighted in **blue**.

Reviewer: 1

The paper introduces EV-GNN, a novel approach leveraging Graph Neural Networks (GNN) and reinforcement learning (RL) to optimize large-scale electric vehicle (EV) charging management. The main goal is to address scalability and uncertainty challenges posed by increasing EV adoption, enhancing energy efficiency.

The paper's major claims are:

- Scalability of EV-GNN: The proposed framework can manage large-scale electric vehicle (EV) charging scenarios effectively by reducing power tracking errors.
- Improved Performance: their results show that EV-GNN outperforms traditional reinforcement learning (RL) methods and heuristic approaches in terms of scalability, energy efficiency, and user satisfaction.
- Innovative Use of GNNs: The combination of Graph Neural Networks (GNNs) and RL offers a novel (and in my honest opinion interesting) way to model the dynamic relationships between vehicles, charging stations, and the electricity grid.

The claims of this paper are novel and relevant in the field of EVs. The integration of GNNs into EV charging management is an innovative application, as this method has rarely been explored in this context as far as I know. Additionally, the work addresses pressing challenges in the EV and energy management community, including scalability and dynamic decision-making in large networks.

Their simulations also integrate real-world data on charging transactions, energy prices, and vehicle characteristics. Their experiments scenarios with small-scale (25 stations) and large-scale (up to 1000 stations) setups were simulated to assess performance and scalability. Simulations are performed on a supercomputer using Nvidia GPUs and high-performance CPUs.

This raises a small problem: simulations require high-performance computing infrastructure, limiting replicability in less-equipped settings. A deeper analysis of computational costs versus the benefits achieved, especially in resource-limited contexts, could add more practical insights.

Authors:

We thank the reviewer for finding our work interesting and novel as we also believe that. We have added the below clarification in the paper (page 11) regarding computation complexity and running on smaller machines:

'RL training was performed on the Delft Blue HPC system~\cite{DHPC2024}, which enabled large-scale simulations with up to 1000 EV chargers and allowed us to explore substantially larger state-action spaces. For our most intensive experiments, we required around 120 GB of RAM to store the RL algorithms' replay buffer, 2 CPU cores, and a single GPU. Importantly, these simulations can also be run on smaller systems by implementing more efficient memory management or batch processing techniques, such as offloading portions of the replay buffer to a hard drive. This approach optimizes resource utilization and makes high-fidelity simulations accessible even without a full-scale supercomputer, albeit at the cost of additional computational time depending on the specific application.'

Suggestion of possible improvement: An exploration of how variations in key parameters (e.g., number of vehicles, charging rates) affect the performance of EV-GNN could provide deeper insights into its robustness.

Authors:

We thank the reviewer for highlighting the robustness of our approach. To validate this robustness, we conducted extensive experiments across various parameter settings and optimization problems. For instance, in the "Generalization to Unseen Environments" section (page 7), we evaluate the performance of our trained model in environments with significantly different problem settings, such as varied EV arrival and departure time distributions (Figure 4.a). Figure 4.b demonstrates that EV-GNN consistently outperforms the baseline methods under these diverse conditions.

Moreover, we explored different action space configurations, such as shifting from continuous to multi-discrete settings as illustrated in Figure 5, and confirmed that EV-GNN remains effective. Additionally, we extended our analysis to a completely different EV smart charging problem, namely V2G profit maximization, which was originally presented in the supplementary material. We have now integrated the "Application to V2G Profit Maximization" section into the main paper (page 8), thereby reinforcing our claim regarding the robustness and generalizability of EV-GNN.

In conclusion, the paper is well-written, and furthermore I appreciate the diagrams and the result plots (especially Figure 4).

Authors:

We thank the reviewer for the kind words!

Reviewer: 3

The authors have addressed a worthy topic and have proposed an innovative solution. However, a few comments from this reviewer will need to be addressed before the manuscript can be considered ready for publication.

1. A clear, concise and specific statement of the study objective seems to be missing. There exist several sentences that allude to this but are rather vague.

Authors:

We thank the reviewer for highlighting the need for a clear study objective. To enhance the clarity of our objective we added the following sentences at the end of the introduction Section:

“EV-GNN not only enhances the scalability of state-of-the-art RL algorithms but also empowers them to handle the complexity and scale of real-world EV charging scenarios. Unlike conventional methods such as stochastic optimization, MPC, or traditional RL approaches, which often struggle with large-scale problem instances, EV-GNN enables efficient and robust decision-making, opening the door to practical applications in large-scale environments.”

2. The EV part of EV-GNN is really about the application context, not the methodology, right? Is this appropriate?

Authors:

We thank the reviewer for the question. While the EV context sets the application stage for EV-GNN, it also motivates a key methodological advancement. By modeling the EV charging problem as a graph-based MDP and using dynamic, end-to-end GNNs, we directly address challenges like complex network interactions and large state-action spaces, issues that traditional methods often struggle with. This engineering-driven extension not only tackles a real-world problem but also offers a versatile framework with strong benchmark performance, making it well-suited for an engineering journal.

3. Figure 1 caption may be shortened significantly. Material could be placed in the main text.

Authors:

We agree with the reviewer that the caption of Figure 1 is too long therefore we moved the implementation details to Sections “RL for optimal EV charging” and highlighted with blue in the manuscript. The new Figure 1 caption is the following:

“The EV charging optimization problem is modeled as a graph, with nodes representing components (EV, charger, transformer, CPO) and their unique features. The graph is simplified by pruning branches where no actions occur, such as charging stations 3, 4, 6, and 7 with no connected EVs. **Heterogeneous node features are processed through node-specific MLPs to transform them into higher-dimensional, homogeneous node**

embeddings of the same size F_0 . \mathbf{c} . The actor NN consists of L sequential GCN layers with F_l features, which process the graph and reduce each node's feature to 1 for continuous actions or \mathcal{A} for multi-discrete actions. EV node features are then selected and mapped to a fixed-size action vector representing the power injection or absorption. \mathbf{d} . The critic NN concatenates the actor's action features with the node state features, processes them through K sequential GCN layers with F_k features, and then mean pools the high-dimensional node features into a fixed-size graph embedding. This embedding is finally fed into an MLP to compute the Q-value.”

4. All symbols in the expressions and equations need to be defined.

Authors:

We carefully re-read the whole document and properly defined all symbols and expressions.

5. Acronyms must be defined at their first instance of use.

Authors:

We carefully re-read the whole document and properly defined all acronyms, such as DR (demand response), etc.

6. Different notations are used to denote the reward function. Need for consistency.

Authors:

We agree with the reviewer regarding reward function consistency. Therefore, the reward functions presented in Equations 10 and 32 now have the same notation.

7. Is there any reason why the reward function is a single attribute function? Would that produce a really robust solution?

Authors:

We thank the reviewer for highlighting the reward design choice. After close collaboration with CPOs in the Netherlands, we found out that the main objective of CPOs is to fairly split their procured energy to all the connected EVs, while also fully charging the EVs by the time of their departure. This real CPO problem boils down to the optimization problem formulation of Equations 11-24. In particular, CPOs procure energy from the energy markets based on their energy demand forecasts. Forecasting the power needs of the next day is out of the scope of this work hence we assume that we have a good enough power setpoint forecast during the simulation day. Therefore, by minimizing the power setpoint tracking error we are indirectly maximizing user satisfaction, as validated by results in Table 1. We have included the following sentence in the manuscript on page 5:

“As shown in Table 1, decreasing the PST error indirectly enhances user satisfaction. Therefore, user satisfaction was not explicitly included in the reward function in Equation Eq.10.”

8. Stronger evidence of the efficacy of the developed method, will be useful to the reader. What is the baseline? What is the integrity of the baseline?

Authors:

We thank the reviewer for raising this important point. Our study benchmarks EV-GNN against standard state-of-the-art model-free RL algorithms across various scales (see Figure 2). To ensure fairness, we conducted multiple training runs with different random seeds and applied identical hyperparameters (e.g., learning rates, parameter counts) across all methods. Additionally, we compared EV-GNN with the rule-based solutions currently used by CPOs. As shown in Table 1, Figure 4, and Figure 6, these baselines underperform in large-scale scenarios, highlighting the superior efficacy of EV-GNN. Our evaluation is further validated by key metrics that CPOs prioritize, such as energy error and user satisfaction.

Submission ID COMMSENG-24-0719-T r2

“Scalable Reinforcement Learning for Large-Scale Coordination of Electric Vehicles Using Graph Neural Networks”

Stavros Orfanoudakis, Valentin Robu, E. Mauricio Salazar,
Peter Palensky, and Pedro P. Vergara

Dear Editor,

Thank you for completing a second review round of our manuscript. We have carefully addressed each comment and suggestion. Detailed responses to each comment are provided below in blue.

REVIEWERS' COMMENTS:

Reviewer #1 (Remarks to the Author):

I believe that the modifications and additions made following the review are quite satisfactory and have indeed improved the paper.

Authors

We thank the reviewer for the positive and constructive feedback they provided in the last review round.

Reviewer #2 (Remarks to the Author):

Thank you very much for your well-written, detailed, and very interesting manuscript. As a computational person myself, dealing with scalability of reinforcement learning is of utmost importance in my area of expertise. In addition to that, the application domain picked is very well-suited to the methodology; something that is sometimes rare, and it adds value to your contribution.

Authors

We thank the reviewer for their kind words.

Since I am reviewing a second iteration, the manuscript has been significantly improved as far as its presentation and its readability, so my comments are very limited. This also speaks to how well-written and interesting the manuscript is overall. Thank you for this

contribution.

1. In the literature review, the authors have primarily focused on computational studies, omitting some research in transportation science or the nexus of transportation and other infrastructure systems. I would recommend that the authors see some of the recent contributions in scheduling (see, e.g., “Electric vehicle community charging hubs in multi-unit dwellings: Scheduling and techno-economic assessment” by Zheng et al., or “Leveraging electric vehicles as a resiliency solution for residential backup power during outages” by Liu, Vlachokostas, and Kontou).

Authors

We agree with the reviewer that our literature review scope can be enhanced with a short discussion about how EVs can contribute to the greater infrastructure. We added the suggested references in the first paragraph of the introduction (See references 3 and 5).

2. In Equation (3), it appears that the index i takes on values 1 to L ; hence it would need to be written as $i = 1, \dots, L$ rather than with the use of the “belongs to” sign ($i \in 1, \dots, L$).

Authors

We thank the reviewer for noticing this. We updated the text of equation 3.

3. I may have missed that (and I apologize if I did; your results discussion was very comprehensive overall): how do you ensure that your training epochs are enough to reach a steady-state in RL?

Authors

Thank you for highlighting this important aspect. Training an RL agent in our highly stochastic and large-scale environment indeed poses unique challenges in determining convergence and ensuring sufficient training epochs. Due to the inherent variability, such as differing EV arrival patterns, charging characteristics, and varying power limits, each training epoch presents a distinct scenario, making traditional convergence criteria insufficient.

To specifically address this, we selected a diverse set of representative scenarios as evaluation benchmarks. Periodically, after a defined number of training epochs, we assess the RL agent's performance on these fixed evaluation scenarios and calculate the average reward. As illustrated in Figure 6, due to different exploration trajectories caused by stochasticity, the convergence to an optimal policy exhibits variability across training

seeds. This justified our approach of evaluating multiple seeds and averaging the results, as shown explicitly in Figures 6a and 6b.

To directly address your question about ensuring sufficient training epochs: we explicitly monitored the evaluation performance over successive epochs and verified empirically that increasing the number of training epochs beyond our reported settings did not yield improved evaluation outcomes. This practical verification confirmed that the chosen epoch count was sufficient for our RL agent to reach a steady-state performance in the tested scenarios.

We added the following clarification in the sub-Section “Training and parameterization”

“Training epochs were empirically validated through repeated evaluations, confirming that additional epochs did not yield improved performance.”

Thank you again for your fantastic contribution. I look forward to seeing it finalized.